# Sample Space Truncation on Boltzmann Machines

**Mahito Sugiyama**
National Institute of Informatics,
JST PRESTO
mahito@nii.ac.jp

**Koji Tsuda**
The University of Tokyo,
RIKEN AIP, NIMS
tsuda@k.u-tokyo.ac.jp

**Hiroyuki Nakahara**
RIKEN CBS
hiro@brain.riken.jp

## Abstract

We present a lightweight variant of Boltzmann machines via sample space truncation, called a *truncated Boltzmann machine* (TBM), which has not been investigated before while can be naturally introduced from the log-linear model viewpoint. TBMs can alleviate the massive computational cost of exact training of Boltzmann machines that requires exponential time evaluation of expected values and the partition function of the model distribution. To analyze the learnability of TBMs, we theoretically provide *bias-variance decomposition* of the log-linear model using dually flat structure of statistical manifolds.

## 1 Introduction

A *Boltzmann machine* (BM) (Ackley et al., 1985), an instance of the log-linear model (Agresti, 2012) and a representative of *Markov Random Fields* (MRFs) (Kindermann and Snell, 1980), models a joint distribution of multiple binary variables. BMs are a fundamental of deep learning, and have been used for various tasks such as density estimation, missing value imputation, and sampling. However, since the number of possible configurations is *exponential* to the number of variables, its exact training is usually intractable. In particular, exact training of BMs requires exponential time evaluation of the expected values and the partition function of the model distribution. To overcome this issue, approximation techniques have been proposed, for example, Gibbs sampling (Casella and George, 1992; Geman and Geman, 1984) and contrastive divergence (Hinton, 2002; Tieleman, 2008) for evaluation of the expected values and annealed importance sampling (AIS) (Salakhutdinov and Murray, 2008) and tracking (Desjardins et al., 2011) for computing the partition function, while exact training of BMs is still a challenging problem.

We tackle this problem by *truncating* the sample space. We propose *truncated Boltzmann machines* (TBMs), which is a lightweight energy-based model on a sample space adaptively constructed from a given dataset. Its training is efficient; the time complexity is linear to the number of variables and the sample size and quadratic to the number of parameters, hence no approximation technique is needed. Moreover, since TBMs do not have any hidden variables, their training is formulated as convex optimization. Furthermore, we theoretically perform *bias-variance decomposition* of the Kullback–Leibler (KL) divergence using *information geometry* (Amari, 2016). Since TBMs belong to a class of the log-linear model on partially ordered sets (Sugiyama et al., 2016, 2017), the resulting statistical manifold has the *dually flat* structure (Amari, 2001), which allows us to apply Pythagorean theorem to achieve bias-variance decomposition.

## 2 Log-Linear Model

First we introduce the log-linear model, which models discrete distributions. We assume that the domain $S$ of distributions is the set $\{0, 1, 2, \ldots, n\}$ without loss of generality, hence a discrete distribution $P$ over $S$ can be treated as a probability vector $\boldsymbol{p} = (p_1, p_2, \ldots, p_n) \in (0, 1)^n$ as

NeurIPS 2020 workshop on Deep Learning through Information Geometry.

the degree of freedom is $n$. Each probability vector $\boldsymbol{p}$ should satisfy $\boldsymbol{p}^T\mathbf{1} < 1$ so that $\boldsymbol{p}^T\mathbf{1} + p_0 = 1$. To simplify the notation, we always treat each vector as a column vector. Uppercase letters $P, Q, R, \ldots$ denote distributions and lowercase letters $p, q, r, \ldots$ denote the corresponding probability mass functions. We use the notation $[n] = \{1, 2, \ldots, n\}$.

In the *log-linear model* (Agresti, 2012), a discrete distribution $P$ over $S$ represented as a probability vector $\boldsymbol{p} \in (0,1)^n$, is modeled via an $n$-dimensional parameter vector $\boldsymbol{\theta} = (\theta_1, \theta_2, \ldots, \theta_n) \in \mathbb{R}^n$ and a *model matrix* $\mathbf{F} \in \mathbb{R}^{n \times n}$, which encodes a relationship between elements and is usually given beforehand. The assumption here is that $\mathbf{F}$ must be *non-singular* to construct a regular statistical model (Agresti, 2012). The general form of the log-linear model is given as

$$\log \boldsymbol{p} = \mathbf{F}\boldsymbol{\theta} - \boldsymbol{\psi}, \tag{1}$$

where $\log$ is an element-wise operation, $\boldsymbol{\psi} = (\psi(\boldsymbol{\theta}), \ldots, \psi(\boldsymbol{\theta})) \in \mathbb{R}^n$ with $\psi(\boldsymbol{\theta}) = -\log p_0$. Equation (1) is often used as a general form of the log-linear model (Coull and Agresti, 2003). For example, when the model matrix $\mathbf{F}$ is the $n \times n$ identity matrix, we obtain the standard discrete distribution, where there is no interaction between elements in $S$. In such a case, $\theta_i = \log(p_i/p_0)$ for each $i \in [n]$. We also introduce an *expectation parameter* $\boldsymbol{\eta} \in (0,1)^n$ defined as $\boldsymbol{\eta} = \mathbf{F}^T\boldsymbol{p}$. In information geometry, it is well known that $(\boldsymbol{\theta}, \boldsymbol{\eta})$ gives the dual coordinate system of the set of $(n+1)$-dimensional discrete distributions, or a *dually flat manifold* (Amari, 2001, 2016), where $\boldsymbol{\theta}$ and $\boldsymbol{\eta}$ are orthogonal with each other.

The *Boltzmann machine* (BM) (Ackley et al., 1985), a well known energy-based model that treats *combinatorial* interaction between binary variables, is realized as a special case of the log-linear model. For a $d$-dimensional binary vector $\boldsymbol{b} \in \{0,1\}^d$, the fully connected BM, modeled as a graph $G = (V, E)$ with $V = \{1, 2, \ldots, d\}$ and $E = V \times V$, is given as

$$\log p(\boldsymbol{b}) = \lambda_\perp + \sum_{i \in [d]} \lambda_{\{i\}} b_i + \sum_{i \in [d]} \sum_{j \in [d]} \lambda_{\{i,j\}} b_i b_j, \tag{2}$$

with a parameter vector $\boldsymbol{\lambda} = (\lambda_\perp, \lambda_{\{1\}}, \ldots, \lambda_{\{d\}}, \lambda_{\{1,2\}}, \ldots, \lambda_{\{d-1,d\}})$. Higher order BMs are defined similarly (Sejnowski, 1986). To clarify the relationship between Equation (2) and the general form in Equation (1), first we prepare a one-to-one indexing mapping $\iota\colon S = \{0, 1, \ldots, 2^d\} \to \{0,1\}^d$ that gives a natural number to each $d$-dimensional binary vector $\boldsymbol{b} \in \{0,1\}^d$, and we assume that $\iota(0) = (0, 0, \ldots, 0)$. Thus each probability $p_i$ in Equation (1) corresponds to $p(\iota(i))$ in Equation (2). In addition, we use an auxiliary function $\sigma\colon \{0,1\}^d \to 2^{\{1, \ldots, d\}}$ that returns indices of "1" of an input binary vector, e.g., $\sigma((0, 1, 1, 0)) = \{2, 3\}$. Then one can see that the log-linear model in Equation (1) coincides with (2) if we let $p_0 = p((0, \ldots, 0))$, $\psi(\boldsymbol{\theta}) = -\lambda_\perp$, and

$$f_{ij} = \begin{cases} 1 & \text{if } \sigma(\iota(j)) \subseteq \sigma(\iota(i)), \\ 0 & \text{otherwise}, \end{cases} \qquad \theta_i = \begin{cases} \lambda_{\sigma(\iota(i))} & \text{if } |\sigma(\iota(i))| \leq 2, \\ 0 & \text{otherwise}. \end{cases} \tag{3}$$

Therefore combinatorial structure of the binary log-linear model in Equation (2), which is realized as the inclusion relationship $\sigma(\boldsymbol{b}) \subseteq \sigma(\boldsymbol{b}')$ for a pair of binary vectors $\boldsymbol{b}, \boldsymbol{b}' \in \{0,1\}^d$, is encoded as a the binary value of the model matrix $\mathbf{F}$. In addition, the binary log-linear model in Equation (2) implicitly perform regularization using the restriction to $\boldsymbol{\theta}$ such that $\theta_i = 0$ if $|\sigma(\iota(i))| > 2$.

## 3 Truncated Boltzmann Machines

Here we formulate TBMs. As we have shown in the previous section, BMs model joint distributions over $d$ binary variables, where the sample space $S$ is always fixed to $\{0,1\}^d$, resulting in the exponentially larger sample space $|S| = 2^d$. This is problematic as the computational cost of computing $\boldsymbol{\eta}$ and $\psi(\boldsymbol{\theta})$ is $O(2^d)$, which is required to compute the gradient of the KL divergence in each iteration of gradient descent to train BMs.

An interesting observation of the log-linear model is that the fixed sample space $\{0,1\}^d$ is not necessarily, and it is possible to *truncate* unnecessary states from the sample space $S$. Assume that a sample $D \subseteq [n]$ is given as data and a parameter domain $B \subseteq [n]$ is fixed. From the definition of log-linear model in Equation (1), the minimum requirement of $S$ is obtained as

$$S = D \cup B \cup \{0\}. \tag{4}$$

This is why the model matrix $\mathbf{F}$ becomes singular if we choose $S' \subset S$ as a sample space. In contrast, Sugiyama et al. (2017) showed that $\mathbf{F}$ is always regular if $S$ is a partially ordered set and

each entry is given as $f_{ij} = \mathbf{1}_{j \leq i}$, where $i \leq j \iff \sigma(\iota(j)) \subseteq \sigma(\iota(i))$ in our case. Our proposal is to set $S$ in Equation (4) as the sample space of the log-linear model; that is, truncate unnecessary space $\{0,1\}^d \setminus S$, and learn the distribution on the space, where the model matrix is also given by Equation (3). Surprisingly, this truncated sample space has not been investigated before in the literature of Boltzmann machines or Ising models.

Learning is achieved by convex optimization, where the objective is to minimize the KL divergence from the empirical distribution $\hat{P}$ given by $D$ to the model distribution $R$. An algorithm for the first-order optimization (gradient descent) is summarized in Algorithm 1 in Appendix. Since we have $\nabla D_{\mathrm{KL}}(\hat{P}, R) = \boldsymbol{\eta}^R - \boldsymbol{\eta}^{\hat{P}}$ in each iteration, an update formula of a subvector $\boldsymbol{\theta}_B^R = (\theta_i)_{i \in B}$ of $\boldsymbol{\theta}^R$ with respect to $B$ of a current distribution $R$ is given as $\boldsymbol{\theta}_B^R \leftarrow \boldsymbol{\theta}_B^R - \varepsilon(\boldsymbol{\eta}_B^R - \boldsymbol{\eta}_B^{\hat{P}})$, where $\varepsilon \in \mathbb{R}$ is a learning rate. The time complexity of each update in the first order optimization is $O(|S||B|) = O(|D||B| + |B|^2)$ in the worst case as we need to compute $\boldsymbol{\eta}^R = \mathbf{F} \exp(\mathbf{F}_B^T \boldsymbol{\theta}_B^R - \boldsymbol{\psi}^R)$ to obtain the next gradient from the updated $\boldsymbol{\theta}^R$, where $\mathbf{F}_B$ is the $|D \cup B| \times |B|$ matrix composed of columns of $\mathbf{F}$ with respect to $B$.

In the following, we empirically examine the efficiency and the effectiveness of TBMs compared with two representative models: fully visible Boltzmann machines (BMs) and restricted Boltzmann machines (RBMs).

**Environment and Implementation.** We used Amazon Linux AMI release 2018.03 and ran all experiments on 2.50 GHz Intel Xeon Platinum 8175M CPU and 384 GB of memory. All methods, TBMs, BMs, and RBMs, were implemented in C++ and compiled with gcc 4.8.5. All experiments are conducted on R version 3.6.0. We used contrastive divergence with a single step of alternating Gibbs sampling (persistent CD-1) (Hinton, 2002; Tieleman, 2008) in learning of BMs and RBMs. RBMs and BMs use mini-batch with the size 20 in training. Throughout our experiments, the number of iterations were up to $10^4$ for all methods to ensure the convergence.

**Datasets.** We have collected two dataset, MNIST (LeCun et al., 2007), which is a collection of images of handwritten digits and popularly used as a benchmark dataset of image classification, and ret-1 (Zhang et al., 2014), which is a neural spiking dataset and originally obtained by Lefebvre et al. (2008). In the MNIST dataset, each $28 \times 28$ image is represented as a $784$-dimensional vector. Since each image $\boldsymbol{v} \in \mathbb{R}^{784}$ is not originally a binary vector, we binarized each pixel such that $b(v_i) = 1$ if $v_i > 0$ and $b(v_i) = 0$ otherwise.

**Protocol.** We train each model (TBM, RBM, and BM) on a dataset randomly sampled form each class of our datasets. To measure the quality of learned distributions, we extract feature representations from each trained model and apply classification to them. More precisely, first we collect 50 samples with the sample size $N = 100$ from each class of a dataset, hence in total there are 500 samples in MNIST and 350 samples in ret-1. Then we train each model on each sample. For the TBM and the BM, let $(\lambda_{\{1\}}, \ldots, \lambda_{\{d\}})$ be bias parameters and $(\lambda_{\{1,1\}}, \ldots, \lambda_{\{d-1,d\}})$ be weight parameters in Equation (2). To obtain a feature vector representation of a sample from each trained model, we construct $\boldsymbol{f} = (f_1, \ldots, f_d)$ such that each $f_i = |\lambda_{\{i\}}| + \sum_{j \in [d]} |\lambda_{\{i,j\}}|$ since the absolute value of each parameter represents the intensity of the corresponding variable (or variable interaction) in the model. Similarly, for RBMs with weight parameters $(\lambda_{\{1,1\}}, \ldots, \lambda_{\{d,h\}})$ with $h$ hidden variables, we obtain $\boldsymbol{f}$ by $f_i = |\lambda_{\{i\}}| + \sum_{j \in [h]} |\lambda_{\{i,j\}}|$. We varied the number of weight parameters and examined the scalability and performance sensitivity. To do this, weight parameters are randomly chosen and $\lambda_{\{i,j\}} = 0$ if it is not selected. After obtaining such feature representations for all samples, we perform classification for them. We use $k$NN ($k = 10$) and obtained the accuracy via leave-one-out cross validation. Therefore, high classification accuracy means that the quality of the learned representation of a distribution is high.

**Results & Discussion** Results of running time and classification accuracy are plotted in Figure 1(a) and (b). We present average running time of learning for each sample in Figure 1(a). It is clear that TBMs are much faster than the BMs and RBMs. In particular, if the number of parameters is less than 1,000, it is two and four orders of magnitude faster than RBMs and BMs, respectively. The efficiency of TBMs becomes smaller if we use massive parameters ($\sim 10^5$), which is predicted from the complexity analysis that TBMs requires the quadratic complexity and BMs and RBMs require linear complexity, while it is still an order of magnitude faster than BMs. This result means that our sample space truncation enables us to perform efficient distribution learning without any approximation techniques such as contrastive divergence and mini-batch selection.

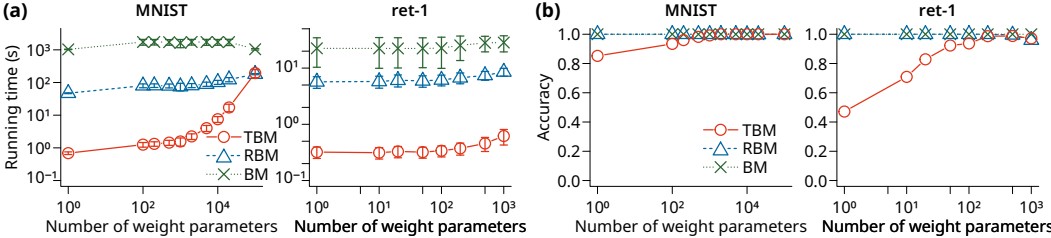

Figure 1: Experimental results of (a) running time (mean±s.d.) and (b) classification accuracy.

Figure 1(b) shows that the classification accuracy of TBMs is almost 1 if we use the reasonable amount of parameters (higher than 100) in both datasets. Hence the loss of information due to sample space truncation can remain at low level, which shows a good trade-off between the effectiveness and the efficiency of TBMs. In addition, we illustrate distribution of feature representations in Figure 3 in Appendix by projecting them onto 2D space by t-SNE (van der Maaten and Hinton, 2008). Classes are clearly separated, which also indicates the effectiveness of TBMs despite the fact that it achieves massive speedup.

### 3.1 Bias-Variance Decomposition

Using the geometric structure of the log-linear model, we analyze its learnability by providing bias-variance decomposition of the KL divergence, which is a fundamental analysis of learning models. In a dually flat manifold, learning of a distribution on $\boldsymbol{S}$ is realized as $e$-projection, which coincides with maximum likelihood estimation (MLE) or KL divergence minimization (Amari, 2016, Chapter 2.8.3). In the following, for each distribution $P$, let $\boldsymbol{\theta}^P$ and $\boldsymbol{\eta}^P$ be its $\boldsymbol{\theta}$- and $\boldsymbol{\eta}$-coordinates, respectively, to clarify that they represent the same distribution $P$. Let $\boldsymbol{S}_{\text{model}}$ be an $e$-flat *model submanifold* given as

$$\boldsymbol{S}_{\text{model}} = \big\{\, P \in \boldsymbol{S} \,\big|\, \theta_i^P = 0 \text{ for all } i \notin B \,\big\} \tag{5}$$

with a predetermined parameter domain $B \in [n]$. Since $\boldsymbol{S}_{\text{model}}$ is determined by the linear constraint on the $\boldsymbol{\theta}$-coordinate, it is a convex set with respect to $\boldsymbol{\theta}$-coordinate. The parameter domain $B$ determines the representation power of the model; overfitting occurs if $B$ is too large and underfitting occurs if $B$ is too small. More precisely, for a series of parameter domains $\emptyset = B_0 \subseteq B_1 \subseteq B_2 \subseteq \cdots \subseteq B_h = [n]$, we obtain the hierarchy of manifolds: $\{P_0\} = \boldsymbol{S}_0 \subseteq \boldsymbol{S}_1 \subseteq \boldsymbol{S}_2 \subseteq \cdots \subseteq \boldsymbol{S}_h = \boldsymbol{S}$, where each $\boldsymbol{S}_j = \{P \in \boldsymbol{S} \mid \theta_i^P = 0 \text{ for all } i \in S \setminus B_j\}$ and $P_0$ is the uniform distribution such that $\boldsymbol{\theta} = \boldsymbol{0}$. For an empirical distribution $\hat{P}$, we obtain an $m$-flat *data submanifold* $\boldsymbol{S}_{\text{data}}$ given as

$$\boldsymbol{S}_{\text{data}} = \left\{\, P \in \boldsymbol{S} \,\Big|\, \eta_i^P = \eta_i^{\hat{P}} \text{ for all } i \in B \,\right\}. \tag{6}$$

Similarly to $\boldsymbol{S}_{\text{model}}$, $\boldsymbol{S}_{\text{data}}$ is a convex set with respect to $\boldsymbol{\eta}$-coordinate. The learning procedure is formulated as a projection of some initial distribution $R \in \boldsymbol{S}_{\text{model}}$ onto the data submanifold $\boldsymbol{S}_{\text{data}}$, which means that it is convex optimization and there exists a unique global solution; that is, the set $\boldsymbol{S}_{\text{model}} \cap \boldsymbol{S}_{\text{data}}$ is always a singleton, thanks to the dually flat structure (Amari, 2009).

Here we perform bias-variance decomposition of the KL-divergence. Our idea is to decompose the expectation of the KL divergence $\mathbb{E}[D_{\text{KL}}(P^*, \hat{P}_B)]$ from the true (unknown) distribution $P^*$ to the MLE (maximum likelihood estimation) $\hat{P}_B$ of an empirical distribution $\hat{P}$ with a fixed parameter domain $B$, and decompose it using the information geometric property. More precisely, let $\boldsymbol{S}_{\text{data}}^*$ be the $m$-flat submanifold that is obtained by replacing $\hat{P}$ with $P^*$ in $\boldsymbol{S}_{\text{data}}$. Since $P^* \in \boldsymbol{S}_{\text{data}}^*$ and $\hat{P}_B \in \boldsymbol{S}_{\text{model}}$, we can apply *orthogonal decomposition* (Amari, 2001) to the KL divergence $D_{\text{KL}}(P^*, \hat{P}_B)$, also known as (generalized) Pythagorean theorem, and obtain the following:

$$\mathbb{E}\Big[D_{\text{KL}}(P^*, \hat{P}_B)\Big] = \underbrace{D_{\text{KL}}(P^*, P_B^*)}_{\text{bias}} + \underbrace{\mathbb{E}\Big[D_{\text{KL}}(P_B^*, \hat{P}_B)\Big]}_{\text{variance}}.$$

The first term corresponds to the bias of the model and measures its representation power, which becomes smaller when the model manifold gets larger. For the variance term, we can lower bound it as $|B|/2N$ using (Barron et al., 1998), which suggests that the variance is *independent of the model structure* **F** even if we have combinatorial structure of variables like BMs or TBMs in it.

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

## A    Information Geometry of Log-Linear Model

We study information geometric structure of the log-linear model. Although more general results are already known in analysis of the exponential family (Amari, 2016) or for graphical models (Wainwright and Jordan, 2008), we explicitly describe it for the better understanding of the geometric structure of the log-linear model.

As we will show in this section, the set of distributions represented by a log-linear model always forms a *dually flat manifold*, which is a statistical manifold with a pair of coordinate systems mainly studied in the area of information geometry (Amari, 2016). A dually flat manifold is generated by a convex function, which is the partition function $\psi(\boldsymbol{\theta})$ in our case. From the condition $\boldsymbol{p}^T\mathbf{1} + p_0 = \boldsymbol{p}^T\mathbf{1} + \exp(-\psi(\boldsymbol{\theta})) = 1$,

$$\psi(\boldsymbol{\theta}) = \log\left(\exp(\mathbf{F}\boldsymbol{\theta})^T\mathbf{1} + 1\right). \tag{7}$$

This is the well-known LogSumExp function and therefore is convex. Let us apply *Legendre transformation* to $\psi(\boldsymbol{\theta})$, which is defined as $\varphi(\boldsymbol{\eta}) = \max_{\boldsymbol{\theta}}(\boldsymbol{\theta}^T\boldsymbol{\eta} - \psi(\boldsymbol{\theta}))$, We have its closed form solution as follows:

**Proposition 1** (Legendre dual). *The Legendre dual $\varphi(\boldsymbol{\eta})$ of $\psi(\boldsymbol{\theta}) = -\log p_0$ is given as*

$$\varphi(\boldsymbol{\eta}) = \log \boldsymbol{p}'^T\boldsymbol{p}', \tag{8}$$

*where $\boldsymbol{p}' = (p_0, \boldsymbol{p}) = (p_0, p_1, \ldots, p_n)$.*

*Proof.* From the Legendre transformation given as

$$\varphi(\boldsymbol{\eta}) = \max_{\boldsymbol{\theta}}\left(\boldsymbol{\theta}^T\boldsymbol{\eta} - \psi(\boldsymbol{\theta})\right), \tag{9}$$

we have

$$\boldsymbol{\theta}^T\boldsymbol{\eta} - \psi(\boldsymbol{\theta}) = \boldsymbol{\theta}^T\boldsymbol{\eta} + \log p_0 = \boldsymbol{\theta}^T\mathbf{F}\boldsymbol{p} + \log p_0 = (\mathbf{F}^T\boldsymbol{\theta})^T\boldsymbol{p} + \log p_0$$
$$= (\log\boldsymbol{p} + \psi(\boldsymbol{\theta}))^T\boldsymbol{p} + \log p_0 = \log\boldsymbol{p}^T\boldsymbol{p} + \psi(\boldsymbol{\theta})\sum\nolimits_{x\in S^+} p(x) + \log p_0$$
$$= \log\boldsymbol{p}^T\boldsymbol{p} + \log p_0\left(1 - \sum\nolimits_{x\in S^+} p(x)\right) = \log\boldsymbol{p}^T\boldsymbol{p} + p_0\log p_0 = \log\boldsymbol{p}'^T\boldsymbol{p}'.$$

Since it follows that $\max_{\boldsymbol{q}'}\log\boldsymbol{q}'^T\boldsymbol{p}' = \log\boldsymbol{p}'^T\boldsymbol{p}'$, Equation (8) holds.    □

Note that we have $\varphi(\boldsymbol{\eta}) = \log(\boldsymbol{F}^{-1}\boldsymbol{\eta}')\boldsymbol{F}^{-1}\boldsymbol{\eta}'$ with $\boldsymbol{\eta}' = (1, \boldsymbol{\eta})$, hence we can see that $\varphi(\boldsymbol{\eta})$ is a function of $\boldsymbol{\eta}$.

Now we consider the geometric structure of the statistical manifold $\boldsymbol{\mathcal{S}} = \{P \mid \boldsymbol{p} \in (0,1)^n, \boldsymbol{p}^T\mathbf{1} < 1\}$, which is the set of $(n+1)$-dimensional discrete distributions. In information geometry, it is known that a pair of coordinate systems is obtained as the gradient of $\psi(\boldsymbol{\theta})$ and $\varphi(\boldsymbol{\eta}))$, respectively (Rockafellar, 1970; Banerjee et al., 2005). As we show in the following proposition, the coordinates interestingly coincide with $\boldsymbol{\theta}$ and $\boldsymbol{\eta}$.

**Proposition 2** (dual coordinate). *For the partition function $\psi(\boldsymbol{\theta})$ of the log-linear model in Equation (1) and its Legendre dual $\varphi(\boldsymbol{\eta})$ given in Equation (8), the dual coordinate system of the statistical manifold $\boldsymbol{\mathcal{S}}$ is obtained as*

$$\nabla\varphi(\boldsymbol{\eta}) = \boldsymbol{\theta}, \quad \nabla\psi(\boldsymbol{\theta}) = \boldsymbol{\eta}. \tag{10}$$

*Proof.* First we prove that the parameter vector $\boldsymbol{\theta}$ coincides with $\nabla\varphi(\boldsymbol{\eta})$. Since we have

$$\frac{\partial p_j}{\partial \eta_j} = \frac{\partial}{\partial \eta_j} \sum_{k\in[n]} f_{kj}^{-1} \eta_k = f_{ij}^{-1},$$

$$\frac{\partial}{\partial \eta_i} p_0 \log p_0 = \frac{\partial}{\partial \eta_i} \left(1 - \sum_{j\in[n]} p_j\right) \log \left(1 - \sum_{j\in[n]} p_j\right) = -\left(1 + \log p_0\right) \sum_{j\in[n]} f_{ij}^{-1},$$

where $f_{ij}^{-1}$ is the $(i,j)$ entry of $\mathbf{F}^{-1}$, it follows that

$$\frac{\partial\varphi(\boldsymbol{\eta})}{\partial \eta_i} = \frac{\partial}{\partial \eta_i} \left(\sum_{j\in[n]} p_j \log p_j + p_0 \log p_0\right) = \sum_{j\in[n]} f_{ij}^{-1} \log p_j + \sum_{j\in[n]} f_{ij}^{-1} - \left(1 + \log p_0\right) \sum_{j\in[n]} f_{ij}^{-1}$$

$$= \sum_{j\in[n]} f_{ij}^{-1} \log p_j + \sum_{j\in[n]} f_{ij}^{-1} \psi(\boldsymbol{\theta}) = \theta_i.$$

The last equality comes from $\boldsymbol{\theta} = \mathbf{F}^{-1} \log \boldsymbol{p} + \mathbf{F}^{-1} \boldsymbol{\psi}$.

Next, we prove $\boldsymbol{\eta} = \mathbf{F}\boldsymbol{p}$. From Equation (7),

$$\frac{\partial\psi(\boldsymbol{\theta})}{\partial \theta_i} = \frac{1}{\exp(\psi(\boldsymbol{\theta}))} \sum_{k\in[n]} \exp\left(\sum_{j\in[n]} f_{kj}\theta_j\right) f_{ki} = \sum_{k\in[n]} f_{ki} p(x) = \eta_i.$$

Hence Equation (10) follows. $\qquad\square$

The resulting manifold $\boldsymbol{\mathcal{S}}$ with the pair of coordinate systems $(\boldsymbol{\theta}, \boldsymbol{\eta})$ is said to be *dually flat*.

In any dually flat manifold, the Fisher information matrix coincides with Riemannian metric, which is defined as $\nabla\nabla\psi(\boldsymbol{\theta})$ and $\nabla\nabla\varphi(\boldsymbol{\eta})$ for $\boldsymbol{\theta}$- and $\boldsymbol{\eta}$-coordinates, respectively. We firstly obtain the closed form solution of the Riemannian metric in the general case of the log-linear model.

**Theorem 1** (Riemannian metric). *Riemannian metric for $\boldsymbol{\theta}$- and $\boldsymbol{\eta}$-coordinates $g_{ij}(\boldsymbol{\theta})$ and $g_{ij}(\boldsymbol{\eta})$ of the log-linear model are obtained as*

$$g_{ij}(\boldsymbol{\theta}) = \sum_{k\in[n]} f_{ki}f_{kj}p_k - \eta_i\eta_j, \tag{11}$$

$$g_{ij}(\boldsymbol{\eta}) = \sum_{k\in[n]} f_{ik}^{-1}f_{jk}^{-1}\frac{1}{p_k} + \sum_{k\in[n]}\sum_{l\in[n]} f_{ik}^{-1}f_{jl}^{-1}\frac{1}{p_0}. \tag{12}$$

*Proof.* In the log-linear model, the Riemannian metric for the $\boldsymbol{\theta}$-coordinate is obtained as

$$\frac{\partial^2}{\partial \theta_i \partial \theta_j}\psi(\boldsymbol{\theta}) = \frac{\partial \eta_i}{\partial \theta_j} = \frac{\partial}{\partial \theta_j} \sum_{k\in[n]} f_{ki}p_k = \frac{\partial}{\partial \theta_j} \sum_{k\in[n]} f_{ki} \exp\left(\sum_{l\in[n]} f_{kl}\theta_l - \psi(\boldsymbol{\theta})\right)$$

$$= \sum_{k\in[n]} f_{ki} \exp\left(\sum_{l\in[n]} f_{kl}\theta_l - \psi(\boldsymbol{\theta})\right) (f_{kj} - \eta_j)$$

$$= \sum_{k\in[n]} f_{ki}f_{kj}p_k - \eta_j \sum_{k\in[n]} f_{ki}p_k = \sum_{k\in[n]} f_{ki}f_{kj}p_k - \eta_i\eta_j.$$

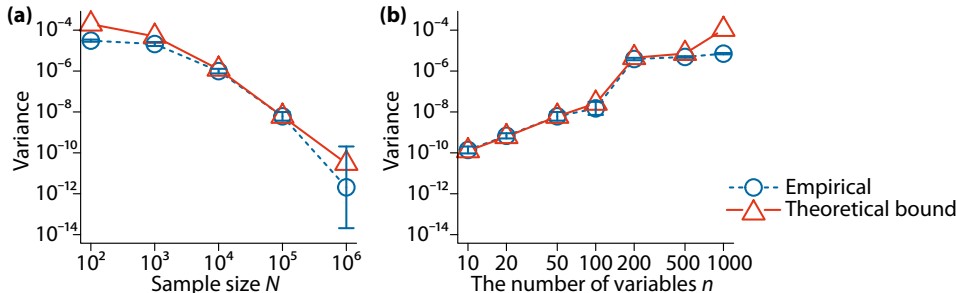

Figure 2: Empirical evaluation of variance. Empirically estimated variances (blue, dotted lines) and theoretically obtained lower bounds (red, solid lines) for (a) $n = 50$ and (b) $N = 100,000$.

The Riemannian metric for $\boldsymbol{\eta}$-coordinate is obtained as

$$\frac{\partial^2}{\partial \eta_i \partial \eta_j} \varphi(\boldsymbol{\eta}) = \frac{\partial \theta_i}{\partial \eta_j} = \frac{\partial}{\partial \eta_j} \left( \sum_{k \in [n]} f_{ik}^{-1} \log p_k + \sum_{k \in [n]} f_{ik}^{-1} \psi(\boldsymbol{\theta}) \right),$$

where we have

$$\frac{\partial}{\partial \eta_j} \sum_{k \in [n]} f_{ik}^{-1} \log p_k = \frac{\partial}{\partial \eta_j} \left( \sum_{k \in [n]} f_{ik}^{-1} \log \left( \sum_{l \in [n]} f_{lk} \eta_l \right) \right) = \sum_{k \in [n]} f_{ik}^{-1} f_{jk}^{-1} \frac{1}{p_k},$$

$$\frac{\partial}{\partial \eta_j} \sum_{k \in [n]} f_{ik}^{-1} \psi(\boldsymbol{\theta}) = -\frac{\partial}{\partial \eta_j} \sum_{k \in [n]} f_{ik}^{-1} \log \left( 1 - \sum_{l \in [n]} p_l \right)$$

$$= -\frac{\partial}{\partial \eta_j} \sum_{k \in [n]} f_{ik}^{-1} \log \left( 1 - \sum_{l \in [n]} \sum_{l' \in [n]} f_{l'l}^{-1} \eta_{l'} \right) = \sum_{k \in [n]} \sum_{l \in [n]} f_{ik}^{-1} f_{jl}^{-1} \frac{1}{p_0}.$$

Hence it follows that

$$\frac{\partial^2}{\partial \eta_i \partial \eta_j} \varphi(\boldsymbol{\eta}) = \frac{\partial \theta_i}{\partial \eta_j} = \sum_{k \in [n]} f_{ik}^{-1} f_{jk}^{-1} \frac{1}{p_k} + \sum_{k \in [n]} \sum_{l \in [n]} f_{ik}^{-1} f_{jl}^{-1} \frac{1}{p_0},$$

yielding Equation (12). $\qquad \square$

Moreover, two coordinates $\boldsymbol{\theta}$ and $\boldsymbol{\eta}$ are *orthogonal* with each other, that is,

$$\frac{\partial^2}{\partial \theta_i \partial \eta_j} \psi(\boldsymbol{\theta}) = \frac{\partial \eta_i}{\partial \eta_j} = \delta_{ij}, \qquad \frac{\partial^2}{\partial \eta_i \partial \theta_j} \varphi(\boldsymbol{\eta}) = \frac{\partial \theta_i}{\partial \theta_j} = \delta_{ij},$$

where $\delta_{ij}$ is the Kronecker delta and $\delta_{ij} = 1$ if $i = j$ and $\delta_{ij} = 0$ otherwise. Equivalently, we have

$$\mathbb{E}_k \left[ \frac{\partial}{\partial \eta_i} \log p_k \frac{\partial}{\partial \theta_j} \log p_k \right] = \delta_{ij}. \tag{13}$$

## B  Empirical Validation of Variance

We empirically demonstrate the tightness of the lower bound $|B|/2N$ of the variance. To obtain the variance $\operatorname{var}(P_B^*) = \mathbb{E}[D_{\mathrm{KL}}(P_B^*, \hat{P}_B)]$, first we fix a true distribution $P^*$ generated from the uniform distribution with its sample space $S$ with $|S| = 1,000$ and get $P_B^*$ estimated by a TBM with $\sigma = 0.37$ and $k = 2$, which gives a reasonable amount of parameters $B$. Then the lower bound is obtained as $|B|/2N$. In each trial, we repeat 100 times generating a sample $D$ with the size $N$ from $P^*$ and generated $\hat{P}_B$ with fixing $S$ and $B$ to directly estimate the variance ($\pm$ its standard deviation). In Figure 2(a) the sample size $N$ is varied from 100 to 1,000,000 with fixing the number of variables $n = |V| = 50$ while in Figure 2(b) $n$ is varied from 10 to 1,000 with fixing $N = |D| = 100,000$. These plots clearly show that our lower bound is tight enough across all settings. The lower bound exceeds the actual variance in some cases, which is due to the approximation error of the Taylor series expansion or fluctuation of random sampling.

**Algorithm 1:** Learning of TBM.

**1** **Input:** Sample $D$, Parameter domain $B$, learning rate $\varepsilon$;
**2** $S \leftarrow D \cup B \cup \{0\}$;
**3** Construct $\mathbf{F}_B \in \{0,1\}^{|D \cup B| \times |B|}$ using Equation (3);
**4** Compute empirical distribution $\hat{P}$ from $D$;
**5** $\boldsymbol{\eta}_B^{\hat{P}} \leftarrow \mathbf{F}_B^T \hat{\boldsymbol{p}}$;
**6** Initialize $\boldsymbol{\theta}_B^R \in \mathbb{R}^{|B|}$ of $R$ satisfying $R \in \boldsymbol{\mathcal{S}}_{\text{model}}$, e.g., $\boldsymbol{\theta}_B^R = \mathbf{0}$;
**7** **repeat**
**8** $\quad \boldsymbol{r} \leftarrow \exp(\mathbf{F}_B \boldsymbol{\theta}_B^R)$;   // $\boldsymbol{r}$ is unnormalized
**9** $\quad \psi(\boldsymbol{\theta}^R) \leftarrow \log(\boldsymbol{r}^T \mathbf{1} + 1)$;   // partition function from Equation (7)
**10** $\quad \boldsymbol{r} \leftarrow \boldsymbol{r} / \exp(\psi(\boldsymbol{\theta}^R))$;   // $\boldsymbol{r}$ is normalized
**11** $\quad \boldsymbol{\eta}_B^R \leftarrow \mathbf{F}_B^T \boldsymbol{r}$;
**12** $\quad \boldsymbol{\theta}_B^R \leftarrow \boldsymbol{\theta}_B^R - \varepsilon(\boldsymbol{\eta}_B^R - \boldsymbol{\eta}_B^{\hat{P}})$;
**13** **until** *convergence of* $\boldsymbol{\theta}_B^R$;
**14** $\boldsymbol{r} \leftarrow \exp(\mathbf{F}_B \boldsymbol{\theta}_B^R)$;
**15** $\psi(\boldsymbol{\theta}^R) \leftarrow \log(\boldsymbol{r}^T \mathbf{1} + 1)$;
**16** $\boldsymbol{r} \leftarrow \boldsymbol{r} / \exp(\psi(\boldsymbol{\theta}^R))$;
**17** Output $(\boldsymbol{r}, \boldsymbol{\theta}_B^R)$;

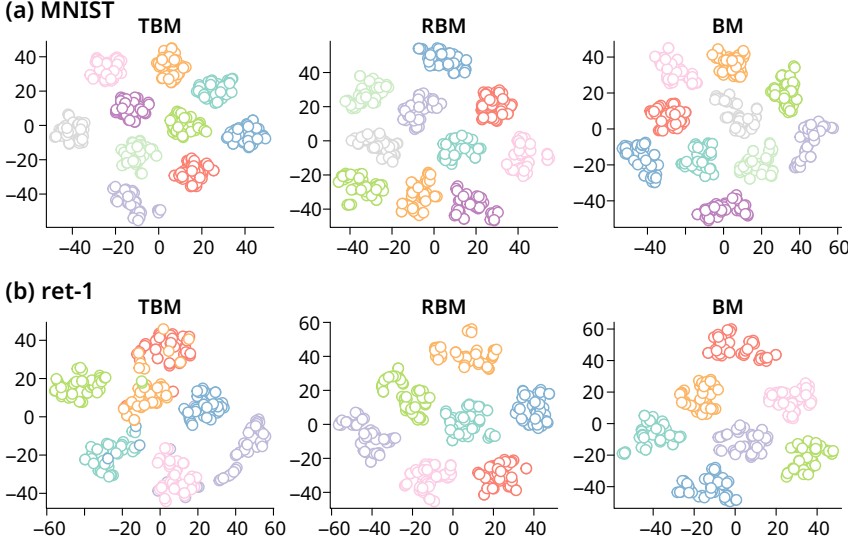

Figure 3: Distribution of feature representations projected on 2D space by tSNE on (a) MNIST and (b) ret-1. Each point corresponds to each sample and colors represents classes. Number of parameters is 1000 for (a) and 100 for (b).

