# OpenReview forum: "Sample Space Truncation on Boltzmann Machines"
_NeurIPS.cc/2020/Workshop/DL-IG — NeurIPSW 2020: DL-IG Poster_

### Official Review · AnonReviewer1 · 2020-10-27
**Efficient learning of BMs (log-linear models) using sample space truncation**

**Rating:** 10
**Confidence:** 4

**Review:**


The paper is well-written and introduce the truncated Boltzmann machines.
First the authors recall the log-models which include the categorical distributions, Boltzmann machine, etc.
Then they notice how sample truncations may improve the learning and describe gradient descent for TBMs.
Finally, thorough experiments are carried out.
An appendix provides a detailed description of the dually flat structure of log-linear models and an empirical study of the variance.


- discuss what happens if p\geq 0 (accepts zero probability for some events), closure

- link of log-linear model with (curved) discrete exponential families?

- an one-to-one indexing -> a one-to-one indexing

- maybe mention equivalent Bregman divergences and KL  (BD as canonical divergences of dually flat spaces)
BD and Pythagorean theorems for the bias-variance decomposition.

- present Fig 1 as 2x2 layout for better reading

- when you say e-projection and KL minimization, precise on which side of KL you minimize

- In the bibliography, missing upper cases, bregman->Bregman, gibbs-> Gibbs, etc.

---

### Official Review · AnonReviewer2 · 2020-11-08
**Review for "Sample Space Truncation on Boltzmann Machines"**

**Rating:** 7
**Confidence:** 3

**Review:**

This paper introduces a modification of traditional Boltzmann machines that restrict the space to training samples which leads to a more tractable optimization for describing an optimal Boltzmann machine on the truncated space. Experiments suggest that this can speed up training with little penalty in performance in some cases, and the bias and variance of the model can be analyzed with information geometry.

Although this model has some interesting theoretical properties, I wonder if I have understood it correctly from the short write-up. It seems that after truncation, it is not possible to evaluate the model on test samples if they appear in the part of the state space that has been truncated. This seems counter-intuitive if true, as most machine learning is concerned with generalization to unseen examples. If I am mistaken in this assertion, perhaps this can be clarified in a longer version of the paper.

---

### Author Response · Authors · 2020-12-11
**Poster**

Poster of this paper is available at:
https://www.dropbox.com/s/7mwpnzt6etxfdqy/Sugiyama_DLIG2020_poster.pdf?dl=0

---

### Decision · Program_Chairs · 2020-11-07

Accept (Poster)